A two-phase transfer learning framework for gastrointestinal diseases classification

Ali Ahmed 1
Iqbal Arshad 1 2 arshad.iqbal@spcai.paf-iast.edu.pk
Khan Sohail 2
Ahmad Naveed 3
Shah Sajid 3
1 School of Computing Sciences, Pak-Austria Fachhochschule: Institute of Applied Sciences and Technology , Mang, Haripur, Khyber Pakhtunkhwa , Pakistan
2 Sino-Pak Center for Artificial Intelligence, Pak-Austria Fachhochschule Institute of Applied Sciences and Technology , Mang, Haripur, Khyber Pakhtunkhwa , Pakistan
3 College of Computer and Information Sciences, Prince Sultan University , Riyadh , Saudi Arabia
Raza Khalid
Electronic publication date: 2024 Dec 19
Publication date: 2024
Volume: 10
Electronic Location ID: e2587
Received 2024 Jul 24; Accepted 2024 Nov 17
Copyright: © 2024 Ali et al.
Copyright year: 2024
Copyright holder: Ali et al.
License: This is an open access article distributed under the terms of the Creative Commons Attribution License, which permits unrestricted use, distribution, reproduction and adaptation in any medium and for any purpose provided that it is properly attributed. For attribution, the original author(s), title, publication source (PeerJ Computer Science) and either DOI or URL of the article must be cited.
License URL: https://creativecommons.org/licenses/by/4.0/

Keywords: Deep learning, Transfer learning, Gastrointestinal (GI) tract, Endoscopic images

Funding: Prince Sultan University The Article Processing Charges (APC) of this publication were financed by Prince Sultan University. The funders had no role in study design, data collection and analysis, decision to publish, or preparation of the manuscript.

==============================
Gastrointestinal (GI) disorders are common and often debilitating health issues that affect a significant portion of the population. Recent advancements in artificial intelligence, particularly computer vision algorithms, have shown great potential in detecting and classifying medical images. These algorithms utilize deep convolutional neural network architectures to learn complex spatial features in images and make predictions for similar unseen images. The proposed study aims to assist gastroenterologists in making more efficient and accurate diagnoses of GI patients by utilizing its two-phase transfer learning framework to identify GI diseases from endoscopic images. Three pre-trained image classification models, namely Xception, InceptionResNetV2, and VGG16, are fine-tuned on publicly available datasets of annotated endoscopic images of the GI tract. Additionally, two custom convolutional neural networks are constructed and fully trained for comparative analysis of their performance. Four different classification tasks are examined based on the endoscopic image categories. The proposed architecture employing InceptionResNetV2 achieves the most consistent and generalized performance across most classification tasks, yielding accuracy scores of 85.7% for general classification of GI tract (eight-category classification), 97.6% for three-diseases classification, 99.5% for polyp identification (binary classification), and 74.2% for binary classification of esophagitis severity on unseen endoscopic images. The results indicate the effectiveness of the two-phase transfer learning framework for clinical use to enhance the identification of GI diseases, aiding in their early diagnosis and treatment.

Introduction

Gastrointestinal (GI) diseases refer to disorders affecting various parts of the GI tract, which play an essential role in the digestive system. Parts of the GI tract include the mouth, throat, gullet, stomach, bowels, rectum, and anal canal, which form the route for all our nutritional intake (Britannica, 2024). The GI tract functions to digest food, take in nutrients, and remove waste from the body. Abnormalities in the GI tract impact the well-being and healthcare of 40–60% of the global population (Sperber et al., 2021). In developed nations, individuals experience a gastrointestinal illness about once every two years, whereas in developing countries, five to ten cases can occur annually (Pierre Payment, 2002). Additionally, over 50,000 new cases of GI illnesses arise every minute, with some patients, particularly young children, losing their lives consequently. Children between the ages of 1 and 4 years have the highest rate of gastrointestinal diseases.

Internal inspection of the GI tract plays an important role in recognizing various GI diseases, particularly those presenting visible mucosa or tissue damage within the GI tract. Gastroenterologists use various non-invasive and invasive techniques to assess the gastrointestinal tract. Non-invasive approaches include imaging methods like ultrasound, CT scans, magnetic resonance imaging (MRI), and X-rays (Frøkjær, Drewes & Gregersen, 2009), along with other functional assessments such as gastric emptying studies, esophageal manometry, pH monitoring, and breath analysis tests (Schwizer, Fox & Steingötter, 2003). Endoscopy is the most common invasive method to examine the GI tract, involving the insertion of light and a camera into the GI tract using a flexible tube through a natural entrance like the mouth. Various endoscopic techniques are used depending on the specific section of the GI tract to be inspected. For instance, esophagogastroduodenoscopy (EGD) examines the upper GI tract, including the esophagus, stomach, and duodenum. Similarly, colonoscopy is used to view the entire large intestine (colon), and sigmoidoscopy is used for inspecting the lower colon and rectum (Banks, 2017). Additional invasive techniques include endoscopic ultrasound (EUS), which combines endoscopy with ultrasound (Fabbri et al., 2014). Capsule endoscopy involves swallowable capsules with cameras to capture images as they travel through the GI tract (Ciuti, Menciassi & Dario, 2011). Endoscopic retrograde cholangiopancreatography (ERCP) merges endoscopy with X-ray imaging to help diagnose and treat various abnormalities (Jorgensen et al., 2016).

Recent developments in computer vision upgrade healthcare diagnostic systems, which assist in automatically detecting various medical conditions, including GI diseases. A key use of computer vision is applying convolutional neural networks (CNNs) to analyze medical images like computed tomography (CT) scans, endoscopic images, magnetic resonance imaging (MRIs), and X-rays. This analysis of medical images helps healthcare specialists in making early diagnoses. Early contributions to computer-aided diagnosis of GI diseases relied on standard approaches, such as feed-forward neural networks to predict conditions like lymph node metastasis in stomach cancer (Hensler et al., 2005). However, with advancements in deep learning methods, GI diagnostic systems are enhanced by integrating CNN-based architectures, such as InceptionNet and VGGNet (Haile et al., 2022). In addition, more recent approaches use attention mechanisms and hybrid methods to address the complexities of large-scale endoscopic data analysis (Lonseko et al., 2021; Fati, Senan & Azar, 2022). Ensemble learning approaches, such as the GastroVRG model proposed by Islam, Rony & Sultan (2024), combine multiple classifiers like VGG, random forest, and gradient boosting to achieve high performance in identifying conditions like esophagitis and polyps.

The proposed study aims to develop a transfer learning framework to train high-performing image classification models for the timely and accurate detection of GI diseases using endoscopic images. This study’s motivation comes from the role of endoscopy in diagnosing GI diseases. However, manually analyzing endoscopic images is a challenging and time-consuming task. It requires extended knowledge and experience in the field of gastroenterology. Additionally, the increasing use of endoscopic procedures and the growing number of images produced add more pressure on healthcare systems. Moreover, a study predicted a shortage of gastroenterology experts by 2020, creating congestion in the diagnostic process and delaying diagnoses for patients in need (Sullivan, 2009). Given these challenges, the motivation is to contribute to developing automated endoscopy systems that could assist gastroenterologists in analyzing endoscopic images more efficiently. Deep CNN models can learn complex visual patterns from image datasets. When integrated with endoscopy systems, these models can automatically identify GI diseases in endoscopic images and reduce the time required for diagnosis. Thus, easing the diagnostic burden on gastroenterologists and ultimately helping them make more informed and timely clinical decisions. Our primary contributions are as follows: A two-phase transfer learning framework is proposed that utilizes pre-trained CNN models and specifically fine-tunes them for clinical use in gastroenterology. Xception, InceptionResNetV2, and VGG16 are employed to classify gastrointestinal diseases using endoscopic images.

The proposed method demonstrates the advantages of fine-tuning pre-trained models, significantly improving consistency and performance. The quantitative results demonstrate notable performance improvements compared to fully trained models.

The proposed framework is flexible and adaptable to various classification tasks in gastroenterology. It can be modified easily for different clinical scenarios by organizing training images into appropriate directories for each new task.

Related work

Many research studies have been conducted on diagnosing gastrointestinal (GI) diseases through machine learning and deep learning. Table 1 provides a summarized overview of related research work. Although many studies have shown encouraging outcomes, there is still a need to enhance the generalization of AI models, which can be achieved by increased diversity in the training data (Niu et al., 2020). Endoscopic data from different gastroenterology departments worldwide should be gathered, and expert gastroenterologists should label the data.

Table 1 Overview of related research work.

Study	Methodology	Results	
Hensler et al. (2005)	Feed-forward neural network aimed at pre-operative detection of lymph node metastasis (LNM) in gastric carcinoma	Accuracy: 72.73%	
Ali et al. (2018)	Gabor-based gray-level co-occurrence matrix (G2LCM) with SVM for chromoendoscopic images	Accuracy: 91%, AUC: 0.87	
Hirasawa et al. (2018)	CNN trained on more than 13,000 endoscopic images for diagnosing gastric diseases	Sensitivity: 92.2%	
Majid et al. (2020)	Wireless Capsule Endoscopy (WCE) system with handcrafted and CNN deep features, feature fusion using a genetic algorithm	Accuracy: 96.5%	
Bang, Cho & Baik (2020)	Fine-tuned InceptionResNetV2 on custom endoscopic dataset for five GI disease categories	Weighted Accuracy: 84.6%	
Yogapriya et al. (2021)	Pre-trained CNNs (VGG-16, ResNet-18, GoogLeNet) fine-tuned for GI image classification	Accuracy: 96.33% (VGG-16)	
Lonseko et al. (2021)	Attention-based CNN for GI disease classification	F1 Score: 0.928, Accuracy: 0.931	
Haile et al. (2022)	InceptionNet and VGGNet features combined, classified using SVM, Random Forest, and KNN	Accuracy: 98% (SVM)	
Fati, Senan & Azar (2022)	Hybrid methods combining CNN-SVM and traditional feature extraction techniques (GLCM, LBP, fuzzy color histogram) for lower GI tumors	Accuracy: 99.3%, AUC: 0.998	
Azar et al. (2023)	CNN-Adam for large intestine cancer classification	Avg. Accuracy: 82%, 95% on specific dataset	
Islam, Rony & Sultan (2024)	Ensemble method (VGG, Random Forest, Gradient Boosting) for esophagitis and polyps classification on Kvasir v1 dataset	Accuracy: 99.73%	
Malik et al. (2024)	Multi-classification deep learning models (VGG-19 + CNN, ResNet152V2, GRU + ResNet152V2, etc.) for WCE image classification	Accuracy: 99.45% (VGG-19 + CNN)	
Mulugeta Abuhayi et al. (2024)	Recurrent Vision Transformer (RVT) with wavelet transform for classifying GI diseases from endoscopic videos	Accuracy: 99.13%, AUC: 0.9954	
Asif et al. (2024)	DCDS-Net combining depth-wise separable convolution with densely connected blocks (DCB) for GI disease classification	Accuracy: 99.33%, Precision: 99.37%, Recall: 99.32%	

Early efforts in computer-aided diagnosis for detecting and classifying GI tract diseases set the basis for recent developments. For example, Hensler et al. (2005) developed a neural network for the pre-operative identification of lymph node metastasis (LNM), a factor used to predict prognosis in stomach cancer. The approach performed better than earlier methods, achieving an accuracy of 72.73%. Building on these foundations, recent approaches use CNNs to classify GI tract diseases. Hirasawa et al. (2018) created a CNN-based diagnostic system trained on over thirteen thousand endoscopic images of GI diseases. A separate dataset of approximately 2,300 endoscopic images is used to test the trained model. The model took less than 50 s to analyze all the images, achieving an overall true-negative rate of 0.922 for polyps in stomach cancer. The study concluded that the proposed model is clinically relevant to assist gastroenterologists in their daily practice. Haile et al. (2022) proposed a technique that combines features extracted from InceptionNet and VGGNet convolutional networks. These combined features are classified using various techniques, including softmax function, random forest, KNN, and SVM. Results show that SVM achieved the highest classification accuracy of 98%.

Further advancements in this field focus on analyzing large volumes of endoscopic video data. Fati, Senan & Azar (2022) worked on the early detection of lower GI tract tumors using endoscopic videos. The study highlights the challenges in analyzing a large number of frames produced in endoscopic videos and suggests using artificial intelligence to help make accurate diagnoses. Four systems utilizing various approaches, including feed-forward and artificial neural networks, pre-trained models like AlexNet and GoogLeNet, hybrid techniques combining CNN models with SVM, and hybrid feature extraction methods incorporating traditional algorithms such as the gray-level co-occurrence matrix, local binary pattern, and fuzzy color histogram are presented. The proposed solution shows better performance in diagnosing tumors in endoscopic video frames. The feed-forward neural network utilizing hybrid features achieves an outstanding accuracy score of 0.993 and an AUC of 0.998.

Attention mechanisms in CNN architectures also show promising results. Lonseko et al. (2021) proposed a deep CNN utilizing attention mechanism for GI disease classification, implemented with encoder and decoder layers. This mechanism performs better than earlier classification methods, which often miss complex spatial dependencies in images due to their structural limitations. The attention model achieves an accuracy of 93.1% and F1-score of 0.928.

Several studies have also explored other feature extraction and optimization methods for classifying GI diseases. For example, Azar et al. (2023) focuses on classifying large intestine cancer, which is among the deadliest cancers. The study examines six deep-learning optimizers, with empirical results revealing CNN-Adam as the overall top performer, achieving 82% average accuracy across four large intestine cancer datasets. In one particular dataset (Dataset-1), CNN-Adam achieves an accuracy of 95%, compared to 76% with RMSprop and 96% with Adadelta. Ali et al. (2018) introduced a feature extraction method combining a Gabor-based gray-level co-occurrence matrix (G2LCM) with an SVM classifier to classify GI diseases in chromoendoscopic images. Their model classifies three classes of GI chromoendoscopic images-one normal and two abnormal- and achieves an accuracy of 91%, a sensitivity of 0.91, a specificity of 0.82, and an AUC score of 0.87.

Transfer learning has also shown effective results in enhancing the performance of CNN models for their use in medical image classification. Bang, Cho & Baik (2020) uses a pre-trained InceptionResNetV2 model and fine-tunes it for a custom endoscopic dataset with five classes of GI tract. The model achieves a weighted accuracy score of 0.846. In the prospective validation phase, the model achieves a lower score of 0.764 than the expert endoscopist with a score of 0.876. Similarly, Yogapriya et al. (2021) used pre-trained CNN models and fine-tuned them to classify GI tract diseases. GoogLeNet, VGG16, and ResNet18 are used as base pre-trained models. The VGG16 achieves the best performance with an accuracy score of 96.33%, effectively differentiating among eight classes of endoscopic images from the GI tract.

Ensemble learning approaches have also been used to improve classification performance. Islam, Rony & Sultan (2024) proposed a novel approach called GastroVRG, which helps in the early screening of GI diseases using advanced transfer features. VGG16, random forest, and gradient boosting models are used to extract features for classifying esophagitis and polyps. The gradient boosting model achieves a classification accuracy of 99.73% when tested on the Kvasir-v1 dataset.

The wireless capsule endoscopy (WCE) is complex and time-consuming because it requires manual review of a large number of frames produced. Researchers have worked on developing automated diagnostic systems for recognizing GI abnormalities through WCE. Majid et al. (2020) proposed a system that includes five key steps: creating a database, extracting handcrafted and CNN deep features, combining features, feature selection by a genetic algorithm, and recognizing diseases. The extracted features include cosine, wavelet, color, and VGG16 feature maps, combined and optimized through a genetic algorithm. An ensemble classifier is then utilized for disease recognition. This technique outperformed existing methods, achieving an accuracy of 96.5% for four types of GI abnormalities, including ulcers, polyps, esophagitis, and bleeding. More recently, Malik et al. (2024) developed four multi-classification deep learning models. The models include VGG19 paired with custom CNN layers, ResNet152V2, gated recurrent unit (GRU) paired with ResNet152V2, and ResNet152V2 paired with bidirectional GRU (Bi-GRU). The models are trained to identify ulcerative colitis, polyps, and dyed-lifted polyps from WCE images. VGG19 paired with custom CNN layers achieved the highest accuracy of 99.45%.

Recent studies have also explored the use of advanced deep learning architectures. Mulugeta Abuhayi et al. (2024) introduced a method that uses recurrent vision Transformers (RVT) combined with wavelet transform to classify GI diseases from endoscopic videos. Their approach includes three key phases: processing images, extracting features, and classification. Wavelet transform is used for segmentation, and adaptive median filtering reduces noise in the image processing phase. Feature extraction is performed using concatenated RVT, and classification is carried out using an ensemble of classifiers, including SVM, Bayesian networks, random forest, and logistic regression. The Hyper-Kvasir dataset is used to test the model. The model achieves an accuracy of 99.13% and an AUC of 0.9954. Another study conducted by Asif et al. (2024) introduced the DCDS-Net model, a deep transfer network that integrates depth-wise separable convolution with densely connected blocks (DCB) and residual connections. The model addresses issues such as overfitting and enhances generalization by integrating Dropout, Batch Normalization, Global Average Pooling, and Dense layers within the DCB. A dataset of 6000 endoscopic images across four classes of GI diseases is used to evaluate the model. DCDS-Net shows high performance, achieving a classification accuracy of 99.33%, a precision of 0.9937, and a recall of 0.9932.

Proposed method

Data acquisition and description

Kvasir-v2 (Pogorelov et al., 2017) is the primary dataset for the proposed research. This dataset was collected from a large gastroenterology center in Norway. Expert endoscopists have labeled and verified the dataset to ensure high-quality annotations. We obtained the dataset from a publicly available repository known as Simula Datasets. There are 8,000 images in the dataset, equally divided into eight categories of the gastrointestinal tract. These include three categories of anatomical landmarks, three categories of pathological findings, and two categories related to polyp removal, described below.

Anatomical landmarks

Anatomical landmarks are visual features in the GI tract that can identify the location of any particular finding for navigation. The three classes of anatomical landmarks are Z-line, pylorus, and cecum. (A) The z-line is a visual boundary between the esophagus and the stomach. During an endoscopy, it appears as a sharp transition from the esophagus’s light mucosa to the stomach’s darker mucosa. (B) The pylorus refers to the region surrounding the opening at the end of the stomach where the small intestine starts. (C) The cecum is pouch-like, and the starting section of the large intestine is located immediately after the small intestine. Figure 1 presents an endoscopic sample for each class in the anatomical landmarks category.

Figure 1 Endoscopic samples of anatomical landmarks, (A) Z-line, (B) pylorus, (C) cecum (Pogorelov et al., 2017).

Pathological findings

Pathological findings are visually affected sections in the GI tract. During an endoscopy, these can be observed as abnormal features from the normal mucosa’s texture, size, and color. These abnormalities either indicate any current disease or the starting of a new disease, possibly leading to cancer. Early detection of pathological findings in a GI patient is crucial for starting treatment timely. The three classes of pathological findings are esophagitis, polyps, and ulcerative colitis. (A) Esophagitis is a noticeable swelling in the esophagus, the tube connecting the mouth to the stomach. This inflammation can be caused by various reasons, such as side-effects of oral medication, allergic reactions, viral/bacterial infections, and conditions like gastroesophageal reflux disease where stomach acid flows backward in the esophagus (Grossi, Ciccaglione & Marzio, 2017). (B) Polyps are growths inside the GI tract that develop into abnormal tissues (Waldum & Fossmark, 2021). Polyps appear in various sizes and shapes; some may indicate early signs of cancer, while most do not develop into cancer (Goddard et al., 2010). (C) Ulcerative colitis is a long-term inflammatory disease in the intestines that leads to swelling and sores in the colon and rectum (Langan et al., 2007). Figure 2 presents an endoscopic sample for each class in the pathological findings category.

Figure 2 Endoscopic samples of pathological findings, (A) esophagitis, (B) polyps, (C) ulcerative colitis (Pogorelov et al., 2017).

Polyp removal

Two classes in the polyp removal category can assist in accurately identifying dyed or lifted polyps for removal and post-removal analysis. (A) Dyed and lifted polyps help identify the location of dyed or lifted polyps, making it easier to separate them from normal tissue during removal. (B) Dyed resection margins are used to determine the post-removal status of the polyps. Figure 3 presents an endoscopic sample for both classes in the polyp category.

Figure 3 Endoscopic samples of polyp removal, (A) dyed and lifted polyps, (B) dyed resection margins (Pogorelov et al., 2017).

Esophagitis severity

In addition to the Kvasir-v2 dataset, a subset of the Hyper-Kvasir dataset (Borgli et al., 2020) is also used. This is to explore a potential clinical application for binary classification of different severity grades of esophagitis disease. The Hyper-Kvasir dataset contains a comprehensive classification of annotated endoscopic images of the GI tract, broadly categorized into upper and lower GI tract. In the upper GI tract, two classes of images are related to the severity of esophagitis. These classes are esophagitis A and esophagitis B-D, as shown in Figs. 4A and 4B. Esophagitis A refers to lower severity of esophagitis, while esophagitis B-D refers to higher severity esophagitis. This system for grading severity is known as the Los Angeles classification system (Lundell et al., 1999), where mucosal breaks under 5 mm are classified as esophagitis A, and those over 5mm fall under esophagitis B, C, or D, with D being the most severe. Automated binary classification between esophagitis A and esophagitis B–D can help differentiate between lower and higher severity levels, supporting early treatment for grade A esophagitis. This data is also obtained from the same repository (Simula Datasets; https://datasets.simula.no/hyper-kvasir/).

Figure 4 Endoscopic samples of esophagitis severity: (A) grade A severity, (B) grades B–D severity (Borgli et al., 2020).

Data preprocessing

This study utilizes the computational resources provided by Google Colaboratory (Bisong, 2019), which offers a cloud-based environment for running Python code, including machine learning tasks. The dataset, available in structured directories, is directly downloaded into the Colab environment for processing. It is first divided into two main parts: 10% is randomly selected and set aside for testing the final trained models, while the remaining 90% is used for training and validation. From this 90%, 70% is allocated for training the model, and the remaining 20% is used for validation, allowing us to monitor and adjust the model’s performance during training. The initial 10% is reserved exclusively for testing the final trained models, ensuring an accurate evaluation of their ability to generalize to new, unseen data.

Three-channel RGB images in the dataset have different dimensions, ranging from (720, 576, 3) to (1,920, 1,072, 3). To ensure uniformity during training, the images are resized to maintain a 1:1 aspect ratio, in line with the input requirements of different model architectures. For example, input images are resized to (299, 299, 3) for the InceptionResNetV2 model and (224, 224, 3) for the VGG16 model. This resizing process ensures that the images are compatible with the different CNN architectures while preserving essential visual information. Figure 5 provides a detailed overview of the end-to-end workflow, highlighting each stage of the process from dataset acquisition to model training, testing, and performance evaluation to facilitate understanding of the entire pipeline. This workflow is followed in all the classification tasks for both the Kvasir and the Hyper-Kvasir datasets.

Figure 5 Overview of the end-to-end workflow.

Transfer learning framework

Transfer learning is a common method in deep learning, particularly in cases where the available dataset is small or specialized. It uses pre-trained models originally optimized for general classification tasks. These models are generally pre-trained on popular datasets, such as ImageNet (Deng et al., 2009) or Microsoft COCO (Lin et al., 2014). Transfer learning adapts the pre-trained models to solve a different classification task. One of the main benefits of transfer learning is that the pre-trained models have already learned important features, and they can be repurposed for a new task, reducing the need for extensive computational resources and data. Several models pre-trained on ImageNet data are employed as the base models for the proposed method. These base models serve as feature extractors for the new task of classifying GI tract endoscopic images. The architecture of these models is customized by removing original classification layers and adding task-specific layers on top of the pre-trained base model. These additional layers are added to meet the specific requirements of the classification task, particularly modifying the final dense layer to match the number of classes and adjusting the activation function based on the environment setup. For example, a softmax activation function is applied in the output layer for multi-class classification. The proposed framework uses a two-phase training process outlined in Algorithm 1.

Algorithm 1 Workflow with the proposed two-phase transfer learning process.

Input: Endoscopic Images	
Output: Trained Models and Evaluations	
Data Partitioning	
testingData← 10% data	
trainingData← 70% data	
validationData← 20% data	
Load Pre-trained Base Model	
numClasses← len(trainingData.classNames)   ▹ number of output classes in a classification task	
baseModel← (keras.applications.PreTrainedCNN:	
  weights = "imagenet"	
  include_top = False)   ▹ pre-trained model with ImageNet weights using Keras applications module. Original classification layer is not included.	
baseModel.trainable←False   ▹ Freeze the layers in the base model	
Add Custom Layers	
x←baseModel(inputs)	
x← GlobalAveragePooling2D(x)	
x← Dropout(0.2)(x) ▹ Add dropout layer to prevent overfitting. 0.2 means that 20% of the nodes will be randomly set to zero during each training epoch.	
outputs← Dense(numClasses, softmax activation)   ▹ Output dense layer automatically gets the number of nodes equal to the number of output classes in the respective classification task	
Compile the Model	
model←inputs, outputs	
optimizer← Adam	
lossFunction← Sparse Categorical Entropy	
learningRate← 0.001	
evaluationMetric← Accuracy	
model.compile← (optimizer, lossFunction, learningRate, evaluationMetric)	
Train Model Custom Layers (Phase 1)	
epochs←50	
model.fit← (trainImages, validImages, epochs)	
Fine-tune the Entire Model (Phase 2)	
baseModel.trainable←True   ▹ Unfreeze base model layers	
learningRate← 0.00001	
model.compile← (optimizer, lossFunction, learningRate, evaluationMetric )   ▹ Recompile the model with a very small learning rate	
epochs←20	
model.fit← (trainImages, validImages, epochs)	
Test the Trained Model	
predictions← model.predict(testImages)   ▹ Test predictions are used to evaluate the model	

Figure 6 demonstrates the architecture of the proposed two-phase transfer learning framework, illustrating how the pre-trained models are integrated and customized for classifying endoscopic images of the GI tract.

Figure 6 Proposed two-phase transfer learning architecture.

1) In the initial training phase, a technique known as “freezing” the pre-trained layers is applied. This means that during backpropagation, only the newly added custom layers are updated, while the weights of the base model remain unchanged. By freezing the pre-trained layers, it is ensured that the model focuses on learning task-specific features from the endoscopic images of the GI tract while still benefiting from the general features (e.g., edges, textures, shapes) learned from the large-scale ImageNet dataset. The custom layers are trained with a higher learning rate to accelerate the convergence of the model’s parameters to the optimal values for the new task.

2) After sufficient training of the custom layers, models are fine-tuned in the second phase. In this phase, the entire model, including the pre-trained layers, is unfrozen and trained on the endoscopic image dataset. Fine-tuning is performed with a very small learning rate (0.00001 for the proposed setup) to ensure that the weights of the pre-trained layers are adjusted carefully, preventing drastic changes that could lead to overfitting or loss of the general features learned from the ImageNet dataset. This phase aims to refine the entire model, allowing the pre-trained layers to adapt to the specific characteristics of the GI tract images while retaining the benefits of transfer learning. This process enhances the model’s ability to generalize to unseen data.

Description of the models used

The above-described transfer learning approach is used to classify endoscopic images of the GI tract, utilizing three well-established pre-trained models. Two custom-designed CNNs are also fully trained for performance comparison. These models are described below.

Pre-trained models

Three CNN models pre-trained on the ImageNet dataset are fine-tuned for the GI tract image classification task. These models—Xception, VGG16, and InceptionResNetV2—are chosen based on their distinct architectures and proven performance in prior transfer learning studies on medical image classification. The Keras applications module is used to import the selected pre-trained models. Table 2 summarizes the modified architectures of the pre-trained models, treating the Xception, VGG16, and InceptionResNetV2 layers as a single base model.

Table 2 Summarized layer-wise architecture of the proposed methodology with pre-trained models.

Model	Layer	Output	Activation	Parameters	
Xception	Input layer	(299, 299, 3)	–	0	
Base model	(10, 10, 2,048)	–	20,861,480	
Average pooling	(1, 1, 2,048)	–	0	
Dropout (0.2)	(1, 1, 2,048)	–	0	
Dense	(1, 1, NC*)	Softmax	∝NC *	
VGG16	Input layer	(224, 224, 3)	–	0	
Base model	(7, 7, 512)	–	14,714,688	
Average pooling	(1, 1, 512)	–	0	
Dropout (0.2)	(1, 1, 512)	–	0	
Dense	(1, 1, NC*)	Softmax	∝NC *	
InceptionResNetV2	Input layer	(299, 299, 3)	–	0	
Base model	(8, 8, 1,536)	–	54,336,736	
Average pooling	(1, 1, 1,536)	–	0	
Dropout (0.2)	(1, 1, 1,536)	–	0	
Dense	(1, 1, NC*)	Softmax	∝NC *	
Note:

* Where NC is the number of classes in a classification task.

Xception (Chollet, 2017), also known as Extreme Inception, is a deep learning model that employs depth-wise separable convolutions to enhance learning efficiency and performance. This model applies depth-wise convolutions to each input channel separately. Depth-wise convolutions are then followed by point-wise convolutions across the input channels. This significantly reduces the number of parameters and computational cost. This design allows for better generalization while maintaining accuracy. The Xception model is fine-tuned to classify endoscopic images of the GI tract by replacing the final fully connected layers with task-specific ones.

VGG16 (Simonyan & Zisserman, 2014) is a popular CNN model in computer vision that has consistently performed well in many image classification tasks, including medical images. It consists of 16 layers: 13 convolutional/pooling layers and three fully connected layers arranged in simple 3 × 3 filter stacks with max-pooling layers between them. Its simplicity makes it a popular choice for transfer learning. The VGG16 model is also fine-tuned for the GI tract image classification tasks.

InceptionResNetV2 combines the multi-scale feature extraction capabilities of the Inception architecture (Szegedy et al., 2017) with the efficiency of residual connections from ResNet (He et al., 2016). The Inception modules simultaneously use convolution filters of varying sizes (1 × 1, 3 × 3, 5 × 5), allowing the model to capture diverse features in the input data. Residual connections help solve vanishing gradient problems by enabling easier learning in deeper networks. The InceptionResNetV2 model is also fine-tuned for the endoscopic image classification tasks.

Each pre-trained model is modified by replacing their original output layers with new fully connected layers suited for various classification tasks. The training process involves freezing the early layers of these models to preserve pre-learned features while fine-tuning the deeper layers to adapt to the endoscopic image dataset of the GI tract. This method maximizes the use of pre-optimized weights from natural images of the ImageNet dataset while ensuring the models are properly tuned to the endoscopic images of the GI tract.

Custom models

Two custom CNN models are also used to compare their performance against three transfer learning models. These models are trained from scratch and used to assess differences in training speed, generalization capability, and the amount of data required for effective training. These models were kept relatively simple to explore how well they could learn spatial features in endoscopic images of the GI tract without the benefit of pre-trained weights. Table 3 summarizes the layered architecture of both custom CNN models.

Table 3 Layer-wise architecture of the custom CNN models.

Model	Layer	Output	Activation	Parameters	
BasicConvNet	Conv2D-1	(299, 299, 75)	Relu	2,100	
MaxPooling2D-1	(150, 150, 75)	–	0	
Conv2D-2	(150, 150, 50)	Relu	33,800	
MaxPooling2D-2	(75, 75, 50)	–	0	
Conv2D-3	(75, 75, 25)	Relu	11,275	
MaxPooling2D-3	(38, 38, 25)	–	0	
Flatten	(1, 1, 36,100)	–	0	
Dense-1	(1, 1, 512)	Relu	18,483,712	
Dense-2	(1, 1, NC*)	Softmax	∝NC *	
RegConvNet	Conv2D-1	(299, 299, 75)	Relu	2,100	
BatchNormalization-1	(299, 299, 75)	–	300	
MaxPooling2D-1	(150, 150, 75)	–	0	
Conv2D-2	(150, 150, 50)	Relu	33,800	
Dropout-1 (0.2)	(150, 150, 50)	–	0	
BatchNormalization-2	(150, 150, 50)	–	200	
MaxPooling2D-2	(75, 75, 50)	–	0	
Conv2D-3	(75, 75, 25)	Relu	11,275	
BatchNormalization-3	(75, 75, 25)	–	100	
MaxPooling2D-3	(38, 38, 25)	–	0	
Flatten	(1, 1, 36,100)	–	0	
Dense-1	(1, 1, 512)	Relu	18,483,712	
Dropout-2 (0.3)	(1, 1, 512)	–	0	
Dense-2	(1, 1, NC*)	Softmax	∝NC *	
Note:

* Where NC is the number of classes in a classification task.

Basic CNN (BasicConvNet) consists of three Conv2D layers, each followed by MaxPooling2D layers and two dense layers. The Conv2D and MaxPooling2D layers extract spatial features from the input images, while the dense layers classify the extracted features. Due to its simplicity, this architecture is prone to overfitting, especially when trained on limited data.

Regularized CNN (RegConvNet) builds on the BasicConvNet structure by incorporating three Conv2D layers, three MaxPooling2D layers, and additional regularization techniques, including BatchNormalization and Dropout layers. The BatchNormalization layers help stabilize and accelerate training by normalizing inputs to each layer, while the dropout layers randomly deactivate a proportion of nodes during training to prevent overfitting. This combination aims to improve generalization on unseen data.

It is important to mention that the BasicConvNet and RegConvNet models do not utilize the two-phase training process associated with transfer learning. Instead, these models are completely trained from scratch, as they lack pre-optimized parameters.

Classification tasks

Throughout the research study, models are trained and evaluated using several classification tasks based on different categories of endoscopic images. This is to explore various scenarios where these models can assist in classifying endoscopic images of the GI tract. These classification tasks are summarized in Table 4.

Table 4 Summary of the classification tasks and their objectives.

Classification task	Dataset	NC *	Objective	
Gastrointestinal tract classification	Kvasir	8	Evaluate the overall performance in classifying GI tract images.	
Pathological findings classification	Kvasir	3	Focus on classifying different GI diseases.	
Polyp identification	Kvasir	2	Distinguish between polyp and normal pylorus classes.	
Esophagitis severity classification	Hyper Kvasir	2	Predict the severity of grade A esophagitis at an early stage.	
Note:

* Where NC is the number of classes in a classification task.

Evaluation method and performance metrics

Trained models are evaluated using a separate testing set of endoscopic images, constituting 10% of the total dataset, which is excluded from the training phase. This helps to assess the models’ generalization capabilities. A model’s performance on unseen data is crucial in determining its effectiveness. Table 5 provides an overview of the testing environment for each classification task.

Table 5 Distribution of test images across four classification tasks.

Classification task	Total images	Images per class	
Gastrointestinal tract classification	800	100	
Pathological findings classification	300	100	
Polyp identification	200	100	
Esophagitis severity classification	66	Esophagitis A: 40	
		Esophagitis B–D: 26	

The following performance metrics are utilized to evaluate models in the proposed method.

Accuracy

Given the balanced class distribution across most classification tasks, accuracy is the primary performance evaluation metric for models in the proposed method. Accuracy measures the ratio of true predictions to the total number of predictions. Mathematically, accuracy for a multi-class classification problem is calculated as:

(1) AAccuracy(%)=∑i=1NCTPi∑i=1NC(TPi+FPi+FNi)×100,

where NC is the total number of classes in a classification task. TPi is the number of true predictions for class i, which refers to the correctly predicted images of that class. FPi is the number of false positives for class i, which represents the total images that were incorrectly classified as class i, but actually belong to a different class. FNi is the number of false negatives for class i, which are the total images that belong to class i but were wrongly classified as another class. For each classification task, three accuracy scores are monitored. Training accuracy indicates the model’s performance on the 70% training set. Validation accuracy reflects the model’s performance on a 20% validation set used during training to tune parameters. Testing accuracy measures models’ performance on the unseen 10% test set.

In addition to accuracy, several statistical performance metrics are considered, particularly for evaluating models in cases of class imbalance in multi-class classification problems. These metrics are weighted average precision WAPrecision, weighted average recall WARecall, and weighted average F1-Score WAF1−Score.

Weighted average precision

WAPrecision measures the precision across all classes, considering class imbalance by weighting each class’s precision by its support. It is defined as:

(2) WAPrecision=∑i=1NC(Pi×Si)∑i=1NCSi

and

(3) Pi=TPiTPi+FPi,

where Pi is the precision for class i, reflecting the proportion of correctly predicted positive instances for that class and offering insight into how well the model avoids false positives. Si represents the support for class i, which is the number of images for class i.

Weighted average recall

WARecall measures the recall across all classes, accounting for class imbalance by weighting each class’s recall by its support. It is defined as:

(4) WARecall=∑i=1NC(Ri×Si)∑i=1NCSi

and

(5) Ri=TPiTPi+FNi,

where Ri is the recall for class i, which is the proportion of true predictions from the actual number of true predictions for that class. It measures how well the model identifies true positive instances for that class.

Weighted average F1-score

WAF1−Score combines precision and recall across all classes, weighted by their support. It is defined as:

(6) WAF1−Score=∑i=1NC(Fi×Si)∑i=1NCSi

and

(7) Fi=2⋅Pi⋅RiPi+Ri,

where Fi is the F1-score for class i, which is the harmonic mean of precision and recall for that class.

Simulation parameters and environment

The proposed method uses the parameters mentioned in Table 6 during the training and fine-tuning processes in a cloud-based Google Colab Python environment using T4 GPU runtime.

Table 6 Parameters used by the proposed method.

Parameter	Value	
Batch size	16	
Optimizer	Adam	
Loss function	Sparse categorical crossentropy	
Learning rate	Phase 1: 0.001	
Phase 2: 0.00001	
Epochs	Phase 1: 50	
Phase 2: 20	

Results

Gastrointestinal tract classification

The first classification task assesses the models’ overall performance in the general classification of the GI tract endoscopic images. In this task, the models are trained to classify eight diverse classes in the Kvasir dataset ( NC=8). In this setup, there are 8,000 images, equally divided among the eight classes, i.e., 1,000 images per class. Following the data partitioning described in Fig. 5, 5,600 images are used to train the models, 1,600 images are used as the validation subset to monitor and optimize the model performance during training and fine-tuning, and the remaining 800 images are used to test the trained models. The classification accuracies of all five models are presented in Table 7. The fine-tuned VGG16 achieves the highest testing accuracy of 87.6%, followed by 85.8% of fine-tuned Xception and 85.7% of fine-tuned InceptionResNetV2, demonstrating adequate performance in classifying unseen endoscopic images from eight different classes. However, the custom CNN models achieve significantly lower testing accuracy scores of 71% by RegConvNet and 60.1% by BasicConvNet. Further results are explored in the discussion section.

Table 7 Accuracy comparison for the gastrointestinal tract classification.

Model	Training	Validation	Testing	
Fine-tuned Xception	96.10%	86.70%	85.80%	
Fine-tuned InceptionResNetV2	91.90%	88.60%	85.70%	
Fine-tuned VGG16	91.20%	87.30%	87.60%	
RegConvNet	95.60%	74.10%	71%	
BasicConvNet	99.30%	64.20%	60.10%	

Pathological findings classification

The second classification task assesses the models to classify three common GI tract diseases, which are the focus of this study. In this task, the models are trained to classify three diseases in the Kvasir dataset ( NC=3), i.e., esophagitis, polyps, and ulcerative colitis. In this setup, there are 3,000 images equally divided among three classes, i.e., 1,000 images for each class. Following the same data partitioning described in Fig. 5, 2,100 instances are utilized for model training, and 600 instances are used as the validation subset to monitor and optimize model performance during the training and fine-tuning phases. As presented in Table 5, the remaining 300 images, completely separated from the training and validation processes, are used to test and evaluate the trained models. The accuracy of the models for disease classification is presented in Table 8. The fine-tuned InceptionResNetV2 achieves the highest testing accuracy of 97.6%, followed by 97.3% of fine-tuned Xception and 96.6% of fine-tuned VGG16, demonstrating nearly excellent performance in classifying unseen endoscopic images of three common diseases of the GI tract. Compared to the previous classification task, the custom CNN models achieve significantly better testing accuracy scores of 87.3% by RegConvNet and 87% by BasicConvNet. Further results are explored in the discussion section.

Table 8 Accuracy comparison for pathological findings classification.

Model	Training	Validation	Testing	
Fine-tuned Xception	99.80%	98.80%	97.30%	
Fine-tuned InceptionResNetV2	98.80%	97.30%	97.60%	
Fine-tuned VGG16	98.30%	96.40%	96.60%	
RegConvNet	99.30%	88.10%	87.30%	
BasicConvNet	100%	86.40%	87%	

Polyp identification

The third classification task assesses the models to perform polyp identification. Models are trained to classify between the normal pylorus and polyps ( NC=2) in the Kvasir dataset. For this task, there are 2,000 instances equally distributed across both classes, i.e., 1,000 per class. Following the same workflow presented in Fig. 5, 1,400 instances are utilized for models’ training and 400 for model validation and optimization. The remaining 200 instances are used as unseen images to test the trained models, as presented in Table 5. Models’ accuracy scores for polyp identification are presented in Table 9. Fine-tuned InceptionResNetV2 achieves the highest testing accuracy of 99.5%, followed by 99% of fine-tuned Xception, 98.5% of BasicConvNet, 97.5% of fine-tuned VGG16, and 95.5% of RegConvNet, demonstrating desirable performance in identifying polyps from normal pylorus by all models. Further results are explored in the discussion section.

Table 9 Accuracy comparison for polyp identification.

Model	Training	Validation	Testing	
Fine-tuned Xception	100%	99.20%	99%	
Fine-tuned InceptionResNetV2	100%	98.90%	99.50%	
Fine-tuned VGG16	100%	98.90%	97.50%	
RegConvNet	99.80%	98.20%	95.50%	
BasicConvNet	99.80%	98.20%	98.50%	

Esophagitis severity classification

The final classification task aims to evaluate the models for their performance in predicting the severity grade of esophagitis disease. The models are trained to perform binary classification among two categories of esophagitis disease in the Hyper-Kvasir dataset ( NC=2), i.e., esophagitis A and esophagitis B–D. There are 663 images distributed unequally among both classes, i.e., 403 instances for esophagitis A and 260 images representing esophagitis B–D. This is the only classification task that presents an imbalance among classes. Again, following the same data partitioning described in Fig. 5, 66 instances are randomly separated for testing the models, 465 instances are utilized for training the models, and 132 are used as the validation subset. As presented in Table 5, out of 66 images in the test set, 40 represent esophagitis A, and 26 represent esophagitis B–D. Table 10 presents the models’ accuracy performance for predicting the severity grade of esophagitis. The results of this task are different from those of all previous tasks. Here, RegConvNet achieves the highest testing accuracy of 77.2%, followed by 74.2% of fine-tuned InceptionResNetV2, 72.6% of fine-tuned Xception, 71.1% of fine-tuned VGG16, and 62.1% of BasicConvNet. All the models show significantly less performance than in previous classification tasks, achieving average classification performance at maximum. Further results are explored in the discussion section.

Table 10 Accuracy comparison for esophagitis severity classification.

Model	Training	Validation	Testing	
Fine-tuned Xception	93.90%	73.40%	72.70%	
Fine-tuned InceptionResNetV2	89.40%	72.70%	74.20%	
Fine-tuned VGG16	89.60%	78%	71.20%	
RegConvNet	99.30%	73.40%	77.20%	
BasicConvNet	72.60%	62.10%	62.10%	

Discussion

For the first classification task, gastrointestinal tract classification, Table 7 results indicate that all five models correctly classify more than 90% of the endoscopic images in the training subset. Classifying endoscopic images in the validation subset, the three fine-tuned models reduce their performance by 5–10%, which is a naturally expected behavior. However, the custom CNN models significantly reduce their performance by more than 20%. This difference between models’ performance on the training images and validation/testing images helps evaluate the generalization behavior of the models. This behavior is known as overfitting when the models show excellent classification performance on the images they were trained on and significantly reduce their performance when tested for unseen images. This behavior is undesirable since it can lead to models’ poor generalization towards unseen data. Figure 7A presents a detailed analysis of models’ training and validation performance for the first classification task throughout the training process. The BasicConvNet exhibits the highest level of overfitting, likely because of its simpler architecture. By adding a few batch normalization and dropout layers to the BasicConvNet architecture, the RegConvNet shows a slight improvement in maintaining higher validation accuracy and reducing overfitting due to these regularization methods. However, it still falls short of ideal performance, particularly for a healthcare classification task with eight categories. The fine-tuned Xception, VGG16, and InceptionResNetV2 models show very similar accuracy scores during training and perform significantly better on unseen images. Another important observation is that the custom CNN models show highly fluctuating performance during training, providing the same learning rate parameter. This is generally caused by insufficient training, and since the custom CNN models are fully trained from scratch, they do not completely learn the underlying patterns from a limited number of endoscopic images. Contrarily, the pre-trained models achieve more consistent performance throughout the training. This benefit is due to their pre-trained parameters and more optimized architecture. The fine-tuning process, depicted in Fig. 7B, further enhances the performance of only pre-trained models. The three fine-tuned models achieve comparable accuracy and generalization performance, with the fine-tuned InceptionResNetV2 showing the best results and the slightest difference between its training and validation accuracies.

Figure 7 Accuracy trends for gastrointestinal tract classification, (A) across training phase, (B) across fine-tuning phase.

For the second classification task, pathological findings classification, Table 8 results indicate that all five models correctly classify more than 95% of the endoscopic images in the training subset. This classification accuracy slightly decreases on the validation and testing subsets by 1–3% for the three fine-tuned models and 10–15% for the custom CNN models. Due to the small number of classes, all five models perform significantly better than the previous classification task. However, the trends and patterns during the training process are quite similar to the previous task, as presented in Fig. 8A. The same overfitting behavior is observed for the BasicConvNet and RegConvNet, with RegConvNet showing even more performance fluctuations. These may also be due to the nonoptimal integration of regularization techniques such as dropout and batch-normalization. The BasicConvNet shows the highest level of overfitting, which shows its inability to generalize well for a three-class classification problem due to an even smaller training dataset. The results also indicate that tuning the hyperparameters of deep learning models can differ across different use cases, even within the same problem area. The transfer learning models demonstrate much higher and comparable performance regarding generalization, achieving an increase of over 10% in accuracy scores compared to the earlier classification task. The InceptionResNetV2 again persists with the highest performance, presenting the same behavior during the fine-tuning process, as presented in Fig. 8B. These results demonstrate that properly fine-tuned transfer learning models are capable and reliable enough to be integrated into modern endoscopy systems for such classification tasks.

Figure 8 Accuracy trends for pathological findings classification, (A) across training phase, (B) across fine-tuning phase.

Moving on to discussing polyp identification results, all five models show excellent accuracy scores for this less challenging binary classification task. Table 9 shows that all five models achieved desirable accuracy scores during training/fine-tuning, validation, and testing. However, looking at Fig. 9A for a detailed analysis, performance fluctuations are extremely elaborated for the RegConvNet. RegConvNet exponentially learns to classify polyps and normal pylorus images during the five training epochs. However, it fluctuates, especially around epochs 25–30, significantly dropping performance. RegConvNet achieves maximum validation and testing accuracy scores of 98.2% and 95.5%, respectively, which are not bad. However, relying upon such models for real-world clinical situations is not recommended, and proper model convergence should be considered as the primary deciding factor. One surprising observation is that the BasicConvNet, the simplest model, achieves better testing accuracy than RegConvNet and even the fine-tuned VGG16. In terms of maintaining performance consistency throughout the fine-tuning phase, Xception shows higher validation scores and slightly excels over the fine-tuned InceptionResNetV2 and fine-tuned VGG16, as presented in Fig. 9B. The fine-tuned InceptionResNetV2 achieves 100% training accuracy and excellent 99.5% testing accuracy, even more than its validation accuracy of 98.9%. This means that out of 200 testing images of polyps and normal pylorus, the fine-tuned InceptionResNetV2 correctly classifies 198 images. This demonstrates the high reliability of a properly fine-tuned InceptionResNetV2 in such clinical scenarios.

Figure 9 Accuracy trends for polyp identification, (A) across training phase, (B) across fine-tuning phase.

For the esophagitis severity classification, results presented in Table 10 indicate overfitting behavior by all five models. Most models, except BasicConvNet, show good training accuracies but significantly reduce their performance on validation and testing images. Contrary to their performance in the previous three classification tasks, all three fine-tuned models also drop their performance by more than 15% during testing. As presented in Fig. 10A, all five models show the most inconsistent performance and achieve the least validation accuracies. Esophagitis severity classification represents a binary classification problem, and the models are expected to perform better. However, they do not. This unexpected behavior appears to be due to the high resemblance between the images of esophagitis A and esophagitis B–D. Let us closely examine the endoscopic samples presented in Fig. 4. We can see that they are quite similar in color and texture features because they come from the same organ, the esophagus. Regarding the fine-tuning process presented in Fig. 10B, all three transfer learning models present better consistency due to a smaller learning rate parameter. The fine-tuned VGG16 slightly excels over the fine-tuned InceptionResNetV2 and fine-tuned Xception. Considering the best performing fine-tuned InceptionResNetV2 with a testing accuracy of 74.2% means that out of 66 unseen images, it correctly classifies around 49 images. This is way less than the desired performance in clinical scenarios, where high misclassification rates lead to high risks in diagnoses and treatments.

Figure 10 Accuracy trends for esophagitis severity classification, (A) across training phase, (B) across fine-tuning phase.

Moreover, the models are evaluated based on other weighted average performance metrics to ensure that primary reliance on accuracy metrics is significant. Table 11 compares weighted average metrics for five models across four classification tasks.

Table 11 Statistical evaluation using weighted average metrics across four classification tasks.

Classification task	Model	WAPrecision	WARecall	WAF1−Score	
Gastrointestinal tract classification	Fine-tuned Xception	0.86	0.86	0.86	
Fine-tuned InceptionResNetV2	0.86	0.86	0.86	
Fine-tuned VGG16	0.88	0.88	0.88	
RegConvNet	0.72	0.71	0.71	
BasicConvNet	0.59	0.60	0.59	
Pathological findings classification	Fine-tuned Xception	0.97	0.97	0.97	
Fine-tuned InceptionResNetV2	0.98	0.98	0.98	
Fine-tuned VGG16	0.97	0.97	0.97	
RegConvNet	0.87	0.87	0.87	
BasicConvNet	0.87	0.87	0.87	
Polyp identification	Fine-tuned Xception	0.99	0.99	0.99	
Fine-tuned InceptionResNetV2	1.00	0.99	0.99	
Fine-tuned VGG16	0.97	0.97	0.97	
RegConvNet	0.96	0.95	0.95	
BasicConvNet	0.99	0.98	0.98	
Esophagitis Severity Classification	Fine-tuned Xception	0.72	0.73	0.72	
Fine-tuned InceptionResNetV2	0.72	0.73	0.73	
Fine-tuned VGG16	0.66	0.67	0.66	
RegConvNet	0.75	0.70	0.70	
BasicConvNet	0.61	0.62	0.55	

It is important to mention that these metrics are obtained from testing the models on unseen images. The first noticeable trend is that the transfer learning models perform better than custom CNN models in most classification tasks. In case of the general classification of endoscopic images of the GI tract in the first task, the fine-tuned VGG16 achieves the highest WAPrecision, WARecall, and WAF1−Score of 0.88, followed by 0.86 of both fine-tuned Xception and fine-tuned InceptionResNetV2. Similar to the accuracy comparison, the custom CNN models are behind the fine-tuned models. In the case of pathological findings classification, the fine-tuned InceptionResNetV2 leads with its excellent weighted average score of 0.98, followed by fine-tuned Xception and fine-tuned VGG16. Here, both custom CNN models achieve same reasonable WAPrecision, WARecall, and WAF1−Score of 0.87. As for the polyp identification, all the models achieve more than 0.95 for all metrics, with the fine-tuned InceptionResNetV2 again leading the chart with WAPrecision of 1.0, WARecall and WAF1−Score of 0.9. Similar to lower accuracy scores, all the models achieve lower weighted average scores of below 0.75 in the case of esophagitis severity classification. The fine-tuned InceptionResNetV2 again achieves the highest scores, WAPrecision of 0.72, WAF1−Score and WARecall of 0.73. The fined-tuned VGG16 shows lower accuracy as compared to the other two fine-tuned models, and performance trends are the same with WAPrecision and WAF1−Score of 0.66, and WARecall of 0.67. These weighted metrics are useful in evaluating models in image classification problems, especially when a class imbalance exists. These metrics are also in line with the accuracy scores, which demonstrate the significance of these results.

In terms of contributing towards automating the diagnoses of GI diseases based on endoscopic images, the proposed study is generally similar to studies cited in the related works table. However, in terms of methods and data, there are certain differences. Some studies use different datasets for GI tract image classification, while others vary in using diverse AI methods. To make a suitable comparison with previous studies, Bang, Cho & Baik (2020) and Yogapriya et al. (2021) are selected based on their relevance for using transfer learning methodology and fine-tuning pre-trained models. Table 12 compares previous studies and two proposed methods. For general classification of endoscopic images of the GI tract, the fine-tuned VGG-16 model by Yogapriya et al. (2021) achieves benchmark performance with an accuracy of 96.33%. A data augmentation strategy is also utilized to expand the number of images in the kvasir-v2 dataset. This increase in data size helps improve the general classification performance of VVG16. This also highlights the effectiveness of using pre-trained CNNs and fine-tuning them for specific medical imaging tasks. Bang, Cho & Baik (2020) fine-tuned the InceptionResNetV2 for five-category classification achieving a weighted accuracy of 84.6%. For specific disease classification, such as classifying a few diseases or identifying polyps from normal pylorus, the proposed fine-tuned InceptionResNetV2 model shows superior performance. The model achieves an accuracy of 97.6% and WAF1−Score of 0.98 for classifying three diseases. For polyp identification, the model achieves an accuracy of 99.5% and WAF1−Score of 0.99, showing the highest and most consistent performance in examined classification tasks. The results suggest that the separate fine-tuning of pre-trained models for specific clinical use cases improves the classification performance. This approach allows the proposed models to capture the intra-variabilities and inter-variabilities of endoscopic images, thus achieving higher performance. The complex architecture of the InceptionResNetV2 model, which integrates both inception modules and residual connections, contributes to its superior performance. The inception modules allow the network to capture multi-scale features, while the residual connections help solve the vanishing gradient problem. The ability of InceptionResNetV2 to consistently achieve higher classification accuracies compared to other models can be attributed to its robust feature extraction capabilities and efficient training dynamics. More importantly, the proposed fine-tuned InceptionResNetV2 model excels in more disease-specific classification tasks. This demonstrates the potential of advanced CNN architectures, when appropriately fine-tuned, to enhance diagnostic accuracy in medical imaging, thereby supporting more reliable and effective clinical decision-making.

Table 12 Comparison with previous studies using transfer learning.

Study	Methodology	Dataset	Results	
Bang, Cho & Baik (2020)	Fine-tuned InceptionResNetV2 on custom endoscopic dataset	Collected 5000+ images for 5 classes	Weighted Accuracy: 84.6%	
Yogapriya et al. (2021)	Pre-trained CNNs (VGG-16, ResNet-18, GoogLeNet) fine-tuned for GI image classification	Kvasir-v2 with 8 classes and data augmentation	Accuracy: 96.33% (VGG-16)	
Proposed	Fine-tuned InceptionResNetV2 for 3 diseases classification	Kvasir-v2 with 3 diseases	Accuracy: 97.6%	
Proposed	Fine-tuned InceptionResNetV2 for Polyp identification	Kvasir-v2 with 2 classes	Accuracy: 99.5%	

Conclusion and future work

Gastrointestinal diseases affect a significant population worldwide. Endoscopy is a common method used by gastroenterologists to examine GI diseases internally. However, manual analysis of a large number of endoscopy images is challenging and time intensive. A two-phase transfer learning framework is proposed to fine-tune pre-trained CNN models to classify endoscopic images of the GI tract. Xception, InceptionResNetV2, and VGG16 have been fine-tuned on Kvasir and Hyper-Kvasir datasets. Two custom CNN models have also been trained for comparative analysis. These models have been tested comprehensively across four classification tasks to explore various clinical scenarios where computer vision can assist in endoscopy. The results show that the transfer learning models perform more consistently in most classification tasks. Transfer learning models also generalize much better due to pre-trained weights, even with small datasets, which is usually a challenge in healthcare imaging. Results also show that these models perform better on specific disease classification tasks than on general classification tasks. Moreover, it is essential to properly plan and optimize regularization techniques, such as dropout and batch normalization, when needed. Improper use of such techniques can negatively impact the model’s performance by increasing the fluctuations and decreasing the convergence during training, leading to an overfit model. In conclusion, the results show that properly fine-tuning pre-trained CNN models can effectively diagnose gastrointestinal tract abnormalities reliably. According to the results, InceptionResNetV2 achieved the highest and most consistent performance across different classification tasks. This shows its potential for use in real-world clinical settings, helping gastroenterologists diagnose and start treatment early to lower mortality rates from gastric diseases. However, there are certain limitations where the clinical scenario of classifying the severity grade of the same disease, like esophagitis, needs to be further explored. For example, the attention and segmentation mechanisms can focus on the size of the infected area since other features, like color, shape, and texture, are mostly the same and not helpful in differentiating lower severity from higher. Furthermore, comprehensive and diverse datasets with high-quality annotated images representing a wide range of GI tract features of different patients would enable further enhanced GI disease detection performance.

Supplemental Information

Supplemental Information 1 Code for A Transfer Learning Framework for Enhanced Detection of Gastrointestinal Tract Pathologies.

Additional Information and Declarations

Competing Interests

Author Contributions

Data Availability

The authors declare that they have no competing interests.

Ahmed Ali conceived and designed the experiments, performed the experiments, analyzed the data, performed the computation work, prepared figures and/or tables, and approved the final draft.

Arshad Iqbal conceived and designed the experiments, performed the experiments, analyzed the data, performed the computation work, prepared figures and/or tables, authored or reviewed drafts of the article, and approved the final draft.

Sohail Khan conceived and designed the experiments, prepared figures and/or tables, authored or reviewed drafts of the article, and approved the final draft.

Naveed Ahmad analyzed the data, authored or reviewed drafts of the article, and approved the final draft.

Sajid Shah analyzed the data, authored or reviewed drafts of the article, and approved the final draft.

The following information was supplied regarding data availability:

Kvasir: A Multi-Class Image-Dataset for Computer Aided Gastrointestinal Disease Detection is available at https://datasets.simula.no/kvasir.

The code for reproducibility is available at GitHub and Zenodo:

- https://github.com/byahmedali/Gastrointestinal-Diseases-Classification.git.

- Ahmed Ali. (2024). byahmedali/Gastrointestinal-Diseases-Classification: A Two-Phase Transfer Learning Framework for Gastrointestinal Diseases Classification (v1.0.0). Zenodo. https://doi.org/10.5281/zenodo.14050606.

- HyperKvasir, a comprehensive multi-class image and video dataset for gastrointestinal endoscopy is available at https://datasets.simula.no/hyper-kvasir/.

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
