# Peer review of "A two-phase transfer learning framework for gastrointestinal diseases classification"

_PeerJ Computer Science, doi:10.7717/peerj-cs.2587_

## Round 0.1 · original submission · Major Revisions

After carefully considering the reviews and assessing your manuscript, I am pleased to inform you that we would like to invite you to revise and resubmit your manuscript for further consideration. The reviewers have provided constructive comments that will help strengthen your work. Please address each of these points thoroughly in your revised manuscript. Additionally, ensure that you provide a detailed response letter outlining how you have addressed each comment raised by the reviewers. This will help the reviewers and myself to evaluate the changes made to the manuscript. It is PeerJ's policy that additional references suggested during the peer-review process should only be included if the authors agree that they are relevant and useful.

·

Basic reporting

Some sections could benefit from further refinement for clarity and precision. While the introduction provides a good context, it would be helpful to expand on certain areas to better establish the motivation behind the study.
Additionally, the literature review should be more comprehensive, particularly in discussing recent advancements relevant to the research topic.

Experimental design

The description of the methods, especially the data preprocessing and model selection, needs more detail to ensure reproducibility.
The authors should provide additional information on the datasets used, including their sources and any preprocessing steps taken. The evaluation metrics and comparison against baseline models should be further elaborated to highlight the significance of the findings.

Validity of the findings

The manuscript could benefit from a more in-depth discussion of the results. The authors should clarify how the findings support the conclusions and address any potential limitations or unresolved questions.
Additionally, the conclusion should be expanded to include future research directions and the broader implications of the work.

Additional comments

Several areas need improvement to enhance the overall quality and impact of the paper. I recommend that the authors address the issues mentioned above and resubmit the revised manuscript for further evaluation.

Reviewer 2 ·

Basic reporting

The manuscript lacks proper English, so it is highly advised to have the proofreading completed by professional services or fluent English support speakers.
Extend the introduction section by adding the latest updates.
Several terms in the Formal results do not include the definitions of the terms used in the equations and theorems.
It is highly recommended that these formulae be updated with proper terms and meanings.
The explanation of each equation should be mentioned; currently, most equations lack it.
Several new citations and references from recent research articles should be included in the manuscript.

Experimental design

The experimental setup needs more elaboration of the terms used in the article for the readers to understand the paper more clearly.
The methodology must be explained in a more detailed way to help the reader understand the parallel recognition approach in the given study.
The effects of the results of the study need to be analyzed in detail.

Validity of the findings

The novelty of research by the authors should be proved by a comparative study.
The authors should add a discussion section. Please add the discussion part before the conclusion section. please mention the key findings of your study.
Highlight the applications and utility of the work.

Reviewer 3 ·

Basic reporting

In recent years, the integration of artificial intelligence (AI) into medical diagnostics has shown immense potential, particularly in enhancing the accuracy and efficiency of disease detection. The proposed framework, "A Transfer Learning Framework for Enhanced Detection of Gastrointestinal Tract Pathologies," seems an interesting research. In this research, the authors leveraging transfer learning techniques to improve the detection of various gastrointestinal (GI) conditions. However, in order to improve the paper quality my suggsetions are given as below.

The title of the paper is not clear. It should be clear and convey spefic meaning. In general, it is not recommended to exceed 10 words.

The authors should add a clear and detailed problem definition.

Authors should give a clear formal definition of the problem.

The authors should add an example to illustrate the problem definition.


The problem is important. it should be mention in the intrdouction section and the abstract section clearly.
The introduction should clearly explain the key limitations of prior work that are relevant to this paper.

Contributions should be highlighted more. It should be made clear what is novel and how it addresses the limitations of prior work.
The role of the introduction is to motivate the problem, briefly highlight limitations or prior work and give a short overview of the contributions.
Novelty is unclear. it should be clear in the revised version.

==== RELATED WORK ====

The related work section is not well organized. Authors must try to categorize the papers and present them in a logical way.

The authors should explain clearly what the differences are between the prior work and the solution presented in this paper.

The authors should add a table that compares the key characteristics of prior work to highlight their differences and limitations. The authors may also consider adding a line in the table to describe the proposed solution.

Experimental design

The experiments should be updated to include some comparison with newer studies.

A statistical analysis should be carried out to demonstrate that the experimental results are significant.

There is not enough discussion of the experimental results.

The experiments have been carried with a few datasets. It is necessary to add more datasets so as to make experiments more convincing.
Pease mention simulation parametrs, enviroment in the result and discussion section. The accuracy measure should be highlighted in this section.
The

Validity of the findings

==== METHOD ====

The authors should first give an overview of their solution before explaining the details.

A novel solution is presented but it is important to better explain the design decisions (e.g. why the solution is designed like that)

It is important to clearly explain what is new and what is not in the proposed solution. If some parts are identical, they should be appropriately cited and differences should be highlighted.

The solution is described but there should be more examples. it is better to add a algorithm.

The authors should add proof(s) of the properties, theorem or lemmas contained in the paper.

The algorithm(s) should be described clearly using pseudocode.
Some text must be added to discuss the future work or research opportunities.

---

## Round 0.2 · accepted · Accept

I am pleased to inform you that your paper has been accepted for publication in PeerJ Computer Science. Your manuscript has undergone rigorous peer review, and I am delighted to say that it has been met with praise from our reviewers and editorial team. Your research makes a significant contribution to the field, and we believe it will be of great interest to our readership. On behalf of the editorial board, I extend our warmest congratulations to you.

Reviewer 2 ·

Basic reporting

I enjoyed reading this manuscript and believe that it is very promising.
Overall, this is an interesting study, and the results obtained are good.

Experimental design

The authors have improved the manuscript as per my previous suggestions. No new suggestions. The manuscript can be considered for acceptance.

Validity of the findings

This method is reliable, rapid, and useful for prediction. There are no new suggestions. The manuscript can be considered for acceptance.

Additional comments

The manuscript has been revised as per my comments and suggestions.